# Recent Progress in Lithium-Ion Battery Safety Monitoring Based on Fiber Bragg Grating Sensors

**DOI:** 10.3390/s23125609

**Published:** 2023-06-15

**Authors:** Dongying Chen, Qiang Zhao, Yi Zheng, Yuzhe Xu, Yonghua Chen, Jiasheng Ni, Yong Zhao

**Affiliations:** 1Institute of Oceanographic Instrumentation, Qilu University of Technology (Shandong Academy of Sciences), Qingdao 266061, China; chendongyingcdy@163.com (D.C.); zhengy@sdas.org (Y.Z.); steny91@gmail.com (Y.X.); zhaoyong@ise.neu.edu.cn (Y.Z.); 2Marine Instrument Center, Pilot National Laboratory for Marine Science and Technology (Qingdao), Qingdao 266237, China; 3Institute of Oceanology, Chinese Academy of Sciences, Qingdao 266071, China; chenyonghua@qdio.ac.cn; 4Laser Institute, Qilu University of Technology (Shandong Academy of Sciences), Jinan 250014, China; njsh51@163.com; 5The College of Information Science and Engineering, Northeastern University, Shenyang 110819, China

**Keywords:** lithium-ion batteries, battery management systems, safety monitoring, fiber Bragg grating sensors

## Abstract

Lithium-ion batteries are widely used in a variety of fields due to their high energy density, high power density, long service life, and environmental friendliness. However, safety accidents with lithium-ion batteries occur frequently. The real-time safety monitoring of lithium-ion batteries is particularly important during their use. The fiber Bragg grating (FBG) sensors have some additional advantages over conventional electrochemical sensors, such as low invasiveness, electromagnetic anti-interference, and insulating properties. This paper reviews lithium-ion battery safety monitoring based on FBG sensors. The principles and sensing performance of FBG sensors are described. The single-parameter monitoring and dual-parameter monitoring of lithium-ion batteries based on FBG sensors are reviewed. The current application state of the monitored data in lithium-ion batteries is summarized. We also present a brief overview of the recent developments in FBG sensors used in lithium-ion batteries. Finally, we discuss future trends in lithium-ion battery safety monitoring based on FBG sensors.

## 1. Introduction

In this paper, we aim to provide a comprehensive analysis of the safety monitoring of lithium-ion batteries based on fiber Bragg grating (FBG) sensors. Our objectives are to explore the potential of FBG sensors in monitoring various parameters, such as temperature, strain, and gas pressure, to enhance the safety, state of charge (SOC), and state of health (SOH) estimation of lithium-ion batteries.

Lithium-ion batteries play a vital role in energy storage devices such as smartphones, laptops, and electric vehicles [1,2]. They provide some advantages, such as a high energy density, environmental friendliness, a longer cycle life [3,4], and so on. The battery management system (BMS) [3,4] has the potential to realize intelligent management and maintenance of each battery cell by preventing overcharge and overdischarge of the battery, extending the service life of the battery, and monitoring the status of the battery. However, current commercial BMS is based on monitoring conventional external parameters such as voltage, current, and external temperature [5,6] and cannot provide adequate information on the internal state in real time. FBG sensors, as a kind of optical fiber sensor, have excellent properties, including microstructure, resistance to electromagnetic perturbations, and distributed measurement capabilities [7,8,9], and can be used in lithium-ion batteries for multiple parameter monitoring.

The lithium-ion batteries heat up and generate strain during normal charging [10,11,12]. Moreover, some abnormal operations, such as overcharging, overpressure, mechanical pressure, and so on, can lead to overheating, deformation, gas generation, and even bulging and thermal runaway [13,14,15]. In the early stages of thermal runaway events, the arising gas causes bulging and increases gas pressure. Therefore, the effective and accurate measurement of temperature, strain, and pressure is helpful to lithium-ion battery safety. Thermocouples or resistance temperature sensors can typically be attached to the surface of batteries to monitor the temperature of lithium-ion batteries [16,17]. However, conventional electrical methods only enable single-point temperature monitoring. Infrared thermography can be used for lithium-ion batteries to obtain spatial temperature distribution [18,19], but it has resolution and accuracy restrictions. The strain on electrodes is mainly measured through the battery’s expansion and contraction [20]. A dilatometer [21], a thickness gauge [22], or X-ray observation [23] can all be used to measure the thickness variation of batteries. However, the indirect measurement technique is unreliable. To accurately define the evolution of electrode strain and enable quick safety warnings, both direct and internal measurements of electrode strain must be carried out. The gas-tight cylinder [24], electronic-type pressure sensors [25], and HM90 high-frequency pressure sensors [26] have been used to measure the gas pressure in lithium-ion batteries. However, these sensors are bulky and cannot perform non-destructive tests. In addition, the liquid electrolyte is an important part of lithium-ion batteries, which can affect the growth of the solid electrolyte interface (SEI) [27,28]. Moreover, the longevity of batteries is greatly influenced by the liquid electrolyte’s stability. Infrared spectroscopy [29], nuclear magnetic resonance [30], and mass spectrometry [31] are a few techniques that have been developed to monitor liquid electrolyte degradation. These methods typically involve post-mortem analysis and are incapable of real-time monitoring. 

The BMS requires an accurate estimation of the SOC and SOH to ensure the safety, life, and performance of the batteries [32,33,34]. In conventional methods, voltage, current, and surface temperature are employed as input parameters to estimate SOC and SOH [35,36,37]. The primary methods for estimating SOC rely on measurements of the batteries’ essential electrical properties, including coulomb counting and open circuit voltage [38,39,40]. However, the electrical-based SOC and SOH estimation methods are vulnerable to electromagnetic interference (EMI) [41,42,43]. External parameters such as the battery voltage, current, and external temperature obtained by the BMS are used for safety warnings [44,45,46]. It was demonstrated that the external parameters of lithium-ion batteries cannot accurately identify thermal runaway because those parameters cannot adequately reflect the internal temperature of the batteries [47]. It was demonstrated that strain monitoring using the FBG sensors can provide earlier warning than conventional temperature monitoring [48]. 

FBG sensors have attracted a lot of interest in lithium-ion batteries due to their excellent properties. The timeline of the development of FBG sensors in lithium-ion batteries is shown in Figure 1. In 2013, Yang et al. [49] employed the FBG sensor to monitor the external temperatures of lithium-ion batteries for the first time. Since then, a variety of parameters of lithium-ion batteries have been monitored using FBG sensors and used to estimate SOC and SOH.

To provide a comprehensive understanding of FBG-based safety monitoring in lithium-ion batteries, we have organized this review as follows: Section 2 will provide an overview of the working principles, fabrication materials, and assembly units of fiber Bragg grating. Section 3 and Section 4 will discuss the single-parameter and dual-parameter monitoring techniques employed in lithium-ion batteries. Furthermore, in Section 5, we will explore the utilization of monitored data. Finally, Section 6 will summarize the conclusions and future perspectives for lithium-ion battery safety monitoring based on FBG sensors.

## 2. Fiber Bragg Grating Sensors

FBG sensors have been widely used in the measurement of multiple quantities. In this section, the sensing principle of fiber Bragg grating sensors, including FBG and tilted fiber Bragg grating (TFBG), is introduced. Fabrication materials and assembly units, two important components of FBG sensors, are also introduced.

### 2.1. Sensing Principle

#### 2.1.1. Sensing Principle of FBG

FBGs can be written in fiber core by changing the core’s refractive index (RI) and forming a periodic modulation along the longitudinal direction. The schematic diagram of the FBG structure is shown in Figure 2. The light propagates in the fiber core and is scattered at each grating surface. The reflection peak will form when the Bragg condition is satisfied. The central wavelength of the reflection peak is affected by the grating parameters. The Bragg condition can be given as [66]
(1)$$\lambda_{B}=2\Lambda n_{eff}$$
where *λ_B_* is the Bragg wavelength, Λ is the grating period that forms the distance between two adjacent grating planes, and *n_eff_* is the effective core RI. The 3 dB bandwidth and the reflection peak height are two important parameters that can directly reflect the FBG performance. To the FBG with a length of 5 mm inscribed by point-by-point inscribing using a femtosecond laser, a 3 dB bandwidth of less than 0.3 nm and a reflection peak height of more than 20 dB can be obtained [67].

The effective core RI and the grating period can vary according to temperature and strain. The wavelength shift induced by temperature and strain can be written as [68]
(2)$$\begin{array}{l} \Delta \lambda_{B}=\lambda_{B}\left(a_{\Lambda }+a_{n}\right)\Delta T+\lambda_{B}\left(1-p_{e}\right)\Delta \varepsilon \\ =k_{T}\Delta T+k_{\varepsilon }\Delta \varepsilon \end{array}$$
where Δ*T* is the temperature change and *ε* is the strain, *a*_Λ_ is the thermal expansion coefficient, *a_n_* is the thermo-optic coefficient, *p_e_* is the strain-optic coefficient, *k_T_* and *k_ε__* are the temperature sensitivity coefficient and strain sensitivity coefficient, respectively, which directly reflect the sensitivity of the FBG sensor. At 1330 nm, the values *a*_Λ_ and *a_n_* are about 0.55 × 10^−6^/°C and 6.6 × 10^−6^/°C, respectively [68]. For germanium-doped silica fiber, the value of *p_e_* is approximately 0.22 [68]. Therefore, the temperature sensitivity coefficient *k_T_* and strain sensitivity coefficient *k_ε_* are 9.51 nm/°C and 1.04 nm/°C, respectively. In practical applications of FBG sensors, high sensitivity, great stability, high linearity, good repeatability, and quick response are required [69,70]. 

As mentioned above, FBG sensors have many advantages. However, they also have some disadvantages. Firstly, FBG sensors are sensitive to multiple parameters, such as temperature, strain, displacement, and pressure. It is not possible to measure every parameter using a single FBG sensor when multiple influencing factors exist at the same time. Secondly, the wavelength demodulation setup, such as the optical spectrum analyzer, is expensive and heavy, and the measurement error is affected by the resolution of the optical spectrum analyzer. Thirdly, FBG sensors are more suitable for measuring static or quasi-static physical quantities (such as temperature, strain, displacement, pressure, etc.) and are not suitable for measuring dynamic signals (such as vibration signals).

#### 2.1.2. Sensing Principle of TFBG

The tilted fiber Bragg grating (TFBG) is sensitive to the surrounding RI and is widely used in biochemical sensing and electrochemical sensing [71]. TFBG can be formed by rotating an angle between the grating plane and the fiber cross-section, as shown in Figure 3. This induces more complex modes of coupling. The forward-propagating core mode couples with the count-propagating core mode and the count-propagating cladding mode. The modes of TFBG include core mode and cladding modes. The resonance wavelength of the core mode is sensitive to multiple physical parameters (temperature, strain, pressure, et al.) and insensitive to the external RI change, which is consistent with that of the FBG. The resonance wavelength of the cladding mode can be expressed as [72]
(3)$$\lambda_{cl}\left(i\right)=\frac{\left(n_{eff,co}+n_{eff,cl}\left(i\right)\right)\Lambda }{\cos\theta }$$
where *λ_cl_*(*i*) is the wavelength of the *i*th cladding mode, *n_eff_*_,*co*_, and *n_eff_*_,*cl*_(*i*) are the effective RI of the fiber core and the *i*th cladding mode, respectively. *θ* is the tilted angle between the grating planes and the cross-section of the fiber. The external refractive index (RI) change can be sensed by the cladding mode of TFBG and induces a shift in the resonance wavelength of the cladding mode [73]. The core mode and cladding molds have the same temperature and strain dependence, and the resonance wavelengths shift together with these parameters. The decoupling of multiple parameters can be achieved by measuring the resonance wavelength shift of core mode and cladding mode. 

### 2.2. Fabrication Materials and Assembly Units

Materials and assembly units are two important parts of advanced FBG sensors that are closely related to manufacturing and cost. They are introduced as follows.

#### 2.2.1. Fabrication Materials

In addition to ordinary silica fibers, FBGs have been successfully fabricated in specially coated fibers, doped fibers, or polymer optical fibers to improve sensing performance and meet environmental needs. To increase the strain sensitivity of FBG, polymer fibers such as polymethylmethacrylate (PMMA), cyclic-olefin copolymer, and polycarbonate fiber have been used instead of conventional silica fiber due to their distinctive material properties. Moreover, FBG can be written in mid-infrared glass fiber, such as fluoride fiber, chalcogenide fiber, and tellurite fiber, which can solve this problem. The temperature sensitivity of chalcogenide (arsenic trifluoride, As2S3) fiber FBG is 175.7 pm/°C, as numerically simulated by Gao et al. [74]. In 2018, Zhang et al. [75] designed a mixed chalcogenide (Ge-Sb-Se)-based multimode FBG, and temperature sensitivity was calculated to be 0.16 nm/°C at 3390 nm. In 2019, Wang et al. [76] designed long-period fiber gratings (LPFGs) in tapered multimode chalcogenide, and the temperature sensitivity can reach 15.2 nm/°C at 1.55 μm, which is about 120 times higher than that of tapered LPFGs made in silica fibers. In 2021, She et al. [77] used a femtosecond laser to fabricate LPFG in fluoride fibers, and the strain sensitivity in the mid-infrared waveband is up to 4.23 pm/με. Compared with traditional silica-based FBGs, the sensitivity performance of these FBGs can be enhanced, but the cost and manufacturing difficulty may increase to a certain extent. 

#### 2.2.2. Assembly Units

Bare FBG sensors without packaging structures are fragile and have low precision. FBG sensor packaging technology can be classified into three kinds in terms of function: protection packaging, sensitization packaging, and compensation packaging. The main packaging methods include tubular packaging, surface-mounted packaging, and polymer-filling packaging. The steel tube, PVC tube, elastic diaphragm, and polymer material are commonly used in FBG sensor packaging structures. The packing of the FBG sensors used in lithium-ion batteries should meet the needs of low material cost, a simple manufacturing process, and ease of manufacture. Peng et al. [78] designed a novel structure to encapsulate the bare FBG, which contains a metal ring and two protective covers. The grating area is loosely attached to the inner groove by epoxy glue to decrease strain cross-sensitivity. The packing FBG sensor can be easily attached to the electrodes of the lithium-ion battery to monitor the temperature. Another packing structure was proposed by them to realize the strain monitoring of lithium-ion batteries, which consists of a sensitivity-enhanced structure and two protective covers [59]. The designed packaging can guarantee easy manufacture, compact size, and minimal strain transmission losses. 

FBG sensors can be attached to the surface or embedded in the interior of the lithium-ion battery to integrate with the battery. They can be used to monitor multiple parameters in different locations of lithium-ion batteries during the overall battery working process. Moreover, they can not only be used for parameter monitoring of a single battery but also for quasi-distributed detections in battery packs. They are proven to be a feasible and non-invasive solution for real-time and multipoint monitoring of lithium-ion batteries, which can improve the BMS and safety of lithium-ion batteries.

## 3. Single-Parameter Monitoring

As mentioned above, FBG sensors have the ability to sense multiple parameters. Temperature and strain are two important parameters that need to be monitored in lithium-ion batteries. In this section, we summarize the research progress on single-parameter (temperature or strain) monitoring of FBG sensors in lithium-ion batteries.

### 3.1. Temperature Monitoring

FBG sensors have been used for temperature monitoring in a variety of lithium-ion batteries, such as cylindrical batteries, pouch batteries, and coin batteries. It is divided into external temperature monitoring and internal temperature monitoring according to the fixed position of FBG sensors.

#### 3.1.1. External Temperature Monitoring

FBG serves as a low-cost thermal diagnostic tool that can be used to monitor the temperature of lithium-ion batteries. Yang et al. [49] pasted FBG sensors on the surface of lithium-ion coin batteries and cylindrical batteries to realize real-time temperature detection. The temperature change of lithium-ion batteries is monitored in real-time by FBG sensors in both normal and abnormal conditions. The FBG sensors show a better thermal response to dynamic loading compared to the thermocouple. In 2017, Nascimento et al. [56] pasted three FBG sensors on the surface of the lithium-ion batteries in real-time to monitor the external temperature changes in three locations of the batteries. As indicated in Figure 4a, three K-type thermocouples were also installed in the same location for comparison testing. The results demonstrated that the FBG sensor surpasses the K-type thermocouple in both resolution and rise time. A FBG sensor is a preferable alternative for monitoring the temperature of lithium-ion batteries in real-time. 

In 2018, a network of 37 FBGs was used to monitor the temperature distribution of three prismatic lithium-ion batteries [57]. As shown in Figure 4b, the 36 FBGs inscribed in four optical fibers were glued to the four interfaces of the lithium-ion batteries in a 3 × 3 matrix. The last FBG sensor maintains a record of the environmental temperature. The thermal mapping of each interface was performed based on the 36 FBG sensors. The charge and discharge cycles of lithium-ion batteries are carried out. There is a noticeable temperature gradient that gradually disappears in the end-of-charge steps. The proposed FBG sensing network can prevent thermal runaway and promote lithium-ion battery safety. In 2021, Peng et al. [78] designed a special FBG encapsulation structure to monitor the electrode temperature, as shown in Figure 4c. The encapsulation structure can be easily attached to the external surface of the electrode of the lithium-ion pouch batteries. The temperature of the lithium-ion batteries was monitored by FBG sensors and PT100 during the charge/discharge cycles, and the experimental data confirmed the consistency of the two methods.

Alcocka et al. [79] attached five FBG sensors to the surface of the cylindrical lithium-ion batteries to monitor the temperature, as shown in Figure 4d. Two mounting techniques for the FBG placed on the lithium-ion are considered. The first mounting method attaches the FBGs to the battery surface with a binding agent. The second mounting method uses a “guide-tube” to reduce longitudinal strain in the FBG sensors. The measurement accuracy of the second method is increased from ±4.25 °C to +2.06 °C. This suggests that the strain may contribute to temperature measurement errors. Although lithium-ion battery external temperature monitoring is simple to use, there are time delays and monitoring errors. 

#### 3.1.2. Internal Temperature Monitoring

FBG sensors are resistant to the harsh chemical environment inside lithium-ion batteries. By implanting the FBG into lithium-ion batteries, it is possible to monitor the internal temperature of such batteries. In 2017, four FBG sensors were used by Novais et al. [54] to monitor the internal and external temperature changes of lithium-ion batteries during the constant current cycle at various C-rates. As presented in Figure 5a, the external sensor is directly attached to the surface of the batteries, while the internal sensor is situated between the two separators. The batteries have a very thin thickness of just 1 mm, and the strain variations are considered null. The temperature sensitivities of the external and internal FBG sensors are about 8.55 pm/°C and 10.24 pm/°C, respectively. During the charging process, the temperature differential between the lithium-ion batteries internal and external temperatures reached 4.7 °C. 

In 2018, Fleming et al. [80] implanted four single-mode FBG sensors into 18650 lithium-ion batteries to realize the quasi-distributed internal temperature measurement of lithium-ion batteries, as illustrated in Figure 5b. The temperature differential between the lithium-ion battery and the tank is significant in charge/discharge cycles, which are 6 °C and 3 °C, respectively. Therefore, it is necessary to monitor the internal temperature of lithium-ion batteries to obtain real-time and accurate temperature data. To assess the maximum charging current, Ametszajew et al. [81] used a FBG sensor to monitor the internal temperature of lithium-ion batteries. 

### 3.2. Strain Monitoring

A FBG sensor can also be fixed on the external surface of batteries or implanted in the electrode of batteries to monitor the strain of lithium-ion batteries.

#### 3.2.1. External Strain Monitoring

In 2019, Peng et al. [59] proposed a sensitivity-enhancing structure to improve the strain sensitivity of lithium-ion batteries. As illustrated in Figure 6a, the sensitization structure includes lever amplification and a flexure hinge. Epoxy glue 353ND was used to join the two ends of FBG1 to the lever mechanism structure. To prevent any strain drift, FBG2 was left loose. The sealed FBG sensors were attached to the surface of the lithium-ion batteries. The strain sensitivity of sealed FBG is 11.55 pm/uε, which is 11.69 times larger than that of bare FBG. The study results demonstrate that the strain increases as the SOC increases at different C-rate charge/discharge cycles.

#### 3.2.2. Internal Strain Monitoring

In 2016, Bae et al. [53] internally implanted a FBG sensor onto the anode of lithium-ion pouch batteries. The anode expansion is significantly larger than the cathode for most lithium-ion batteries. Therefore, the FBG sensor was implanted on the anode of lithium-ion batteries. Two different configurations have been developed: the “attached” method and the “implanted” method, as shown in Figure 6b,c. In the “attached” method, the FBG sensor was embedded between electrodes. In the “implanted” method, the FBG sensor was implanted within the individual anode electrode itself. The peak shifting of the Bragg wavelength was observed in both configuration methods and was caused by the temperature and strain of the anode electrode. The implanted FBG sensor can sense longitudinal and transverse strains induced by an anode electrode, while the attached FBG sensor can only detect the axial strain. In 2022, Blanquer et al. [65] embedded FBG sensors into coin and Swagelok batteries containing either liquid or solid-state electrolytes. The FBG sensor is first injected through two holes with opposite diameters that are perforated inside the body. After the battery is assembled, both sides of the battery are sealed with epoxy resin adhesive to fix the optical fibers at both ends and make the system airtight. During the battery cycle, the optical signal is monitored, further converted into strain, and correlated with the voltage distribution.

In the practical application of lithium-ion batteries, several factors are changed simultaneously in the practical use of lithium-ion batteries. Other parameter changes in lithium-ion batteries need to be isolated to improve the accuracy of single-parameter monitoring.

## 4. Dual-Parameter Monitoring

In lithium-ion batteries, FBG sensors can realize not only single-parameter monitoring but also multi-parameter monitoring. In this section, we summarize the research progress on dual-parameter monitoring of FBG sensors in lithium-ion batteries, such as temperature and strain, temperature and pressure, and temperature and electrolyte RI.

### 4.1. Simultaneous Monitoring of Temperature and Strain

Lithium-ion batteries generate heat and strain during their use. The wavelength shift of the FBG sensor is the result of the interaction of several factors that cannot be separated. In order to implement the simultaneous monitoring of temperature and strain in lithium-ion batteries, many approaches have been proposed.

#### 4.1.1. Parallel Reference FBG

Sommer et al. [50,51] added a parallel reference FBG sensor to realize the simultaneous monitoring of temperature and strain in lithium-ion batteries. Two FBGs inscribed into two different optical fibers were used to simultaneously monitor the temperature and strain of lithium-ion batteries, as seen in Figure 7a. The bonded FBG (FBG1) is tightly attached to the surface of pouch lithium-ion batteries with epoxy adhesive, which is sensitive to both strain and temperature changes. The reference FBG (FBG2) parallel to FBG1 is loosely attached with a heat-conducting paste that is insensitive to strain. The Bragg wavelength shift of two FBG sensors was monitored during charge and discharge of pouch lithium-ion batteries, and the temperature and strain were also obtained, respectively. The test results indicate that the strain curve exhibits distinct characteristics during charge and discharge, and these stage transition points are useful for monitoring lithium-ion batteries. Then, a sensing network composed of five FBG sensors was positioned on the surface of the batteries in the x- and y-directions to monitor the temperature and strain of lithium-ion batteries. The abnormal procedure produced a temperature value of 27.57 ± 0.13 °C near the positive electrode side and strain values of 593.58 ± 0.01 uε along the *y*-axis [58]. Raghavan et al. [52] embedded two parallel FBGs into the electrode of lithium-ion pouch batteries, and the same approach can be used to achieve simultaneous monitoring of temperature and strain.

#### 4.1.2. Tilted and Fixed Three FBGs

Two single-mode FBG sensors placed at a certain angle can be used to decouple the strain and temperature cross-sensitivity of FBG sensors [82]. Three tilted fixed FBG sensors were utilized by Rente et al. [63] to simultaneously monitor the strain and temperature of cylindrical lithium-ion batteries. They were each attached to the surface of 18650 cylindrical batteries at a specific angle, as seen in Figure 7b. The temperature, the radial strain, and the longitudinal strain can all be determined using the layout. The temperature sensitivity and strain sensitivity of the FBG sensors were 14 pm/°C and 1 pm/με, respectively. The radial strain was measured in a series of charge/discharge cycles with a 1C rate.

#### 4.1.3. Polarization-Maintaining FBG

Polarization-maintaining FBG sensors have the ability to realize dual-parameter discrimination [83,84,85]. In 2022, Matuck et al. [64] used PM-FBG sensors to simultaneously monitor the temperature and deformation changes in lithium-ion batteries. As shown in Figure 7c, three PM-FBG sensors inscribed by the phase mask method were fixed in three different zones of the 18650 cylindrical battery surface (anode, middle, and cathode). The simultaneous measurement of temperature and deformation was conducted in two different charge/discharge cycles. The tested data demonstrates that the lowest temperature change and highest deformation occurred in the middle of the lithium-ion batteries in the two charge/discharge cycles.

#### 4.1.4. FBG Combined with FPI

Fabry-Perot Interferometer (FPI) sensors are sensitive to temperature [86], strain [87], pressure [88], RH [89], and so on. Nascimento et al. [60] used a hybrid sensor containing FBGs and FPIs to realize the simultaneous monitoring of temperature and strain of lithium-ion batteries, as shown in Figure 7d. The FBG was inscribed on a single-mode fiber. The FP cavity is manufactured by fusing a multimode fiber in the tail fiber of FBG. The three hybrid sensors were embedded into lithium-ion pouch batteries in three different locations (the top, the middle, and the button). The internal strain and temperature of lithium-ion batteries were monitored during three different steps: constant current (CC) charge, constant voltage (CV) charge, and CC discharge. During the CV charge step, the maximum temperature and strain were observed in the middle of lithium-ion batteries.

### 4.2. Simultaneous Monitoring of Temperature and Pressure

The real-time monitoring of the internal pressure provides an effective pre-warning before gas-release events. Huang et al. [61] achieved the simultaneous monitoring of temperature and pressure using two FBGs inscribed in conventional single-mode fiber (SMF) and microstructure optical fiber (MOF), namely SMF-FBG and MOF-FBG, respectively. The two FBG sensors were bonded on an 18-gauge needle at the same position. As depicted in Figure 8, the needle was placed into the lithium-ion battery jelly roll through a hole that had been bored in the negative electrode. The strain caused by the jelly roll can be avoided with this approach. The battery was then sealed and filled with electrolytes. The two FBGs have varied sensitivities to pressure and temperature. The temperature and pressure were tested during charge/discharge cycles. The FBG sensors have the ability to obtain the key thermodynamic parameters. Thermal incidents can be avoided by configuring the heating/cooling system based on the observed signal.

### 4.3. Simultaneous Monitoring of Temperature and Electrolyte RI

To monitor the temperature and RI of the electrolyte at the same time, a TFBG sensor was implanted into a cylindrical 18650 battery [30]. The excimer laser and phase filter were used to inscribe the TFBG in a single-mode fiber. The sensitivity of TFBG obtained by testing sucrose solution is 6 × 10^−5^ RIU. As described in Ref. [84], TFBG was injected into the Na-ion batteries through a 0.8 mm hole without the interference of pressure. The turbidity of electrolytes was assessed, and the (electro-) chemical reaction pathways were successfully identified by the TFBG.

The single-parameter and dual-parameter monitoring of lithium-ion batteries has been realized based on FBG sensors. The summary of some FBG sensors developed to monitor different parameters of lithium-ion batteries (temperature, strain, pressure, and electrolyte RI) is listed in Table 1.

FBG sensors have been successfully used in different battery forms, such as coin cells, pouch cells, cylindrical cells, and Swagelok cells. In pouch cells and cylindrical cells, the external temperature and strain can be monitored by FBG sensors. The temperature and strain of their packings can be easily monitored by pasting the FBG sensors on the surface of the battery packings. The monitored data can be used by the BMS to provide battery state estimation. However, the electrode strain of coin cells and Swagelok (including solid-state batteries) cells is difficult to test externally due to their hard casings. The FBG sensors needed to be inserted into the two kinds of batteries by making slots in the casing, which makes quasi-distributed monitoring of their battery packs difficult. So FBG sensors are more suitable for the parameter monitoring of a single coin cell or Swagelok cell. In the future, FBG sensors are expected to be used in wearable batteries, but external environmental factors such as bending and vibration need to be considered.

## 5. Utilization of Monitored Data

The aim of battery sense is to help the BMS control the energy flow in and out of the battery, ensuring its safety, and optimizing the use of the energy inside the battery. In this section, we summarize the utilization of the monitored data, including the estimation of SOC and SOH and safety warnings.

### 5.1. SOC and SOH Estimation of Lithium-Ion Batteries

The wavelength shifting of FBG as a function of SOC and SOH was observed. SOC and SOH can be estimated based on the temperature and strain of lithium-ion batteries. The main methods are summarized below.

#### 5.1.1. Dynamic Time-Warping Algorithm

The Dynamic Time Warping (DTW) algorithm, as a machine learning algorithm, can be used to measure the similarity between two temporal sequences changing at various speeds [90,91]. Ganguli et al. [92] estimated the SOC of lithium-ion batteries using the dynamic time-warping (DTW) algorithm and Kalman filtering based on the internal electrode strain of pouch batteries. It was shown that SOC estimation accuracy is better than 2.5% under different temperature conditions and dynamic cycling. SOH estimation capacity up to 5 cycles with an inaccuracy of roughly 1.1% or less. They used a linear Kalman filtering algorithm to predict capacity, and the error is less than 2%. A SOC estimation approach combining the surface strain of cylindrical batteries with DTW algorithms was proposed by Rente et al. [63]. The resolution and accuracy of SOC estimation are 1% and 2%, respectively.

#### 5.1.2. Deep Neural Network

The deep neural network (DNN) is one kind of artificial neural network that has strong nonlinear predictive ability [93]. The relationship between SOC and strain is extremely complicated and cannot be described just by a simple polynomial function. Therefore, DNN can be used as a tool for determining the SOC. Ee et al. [62] introduced the SOC estimation approach based on nonelectrical parameters using DNN. The surface strain, temperature, and strain rate are the input parameters for the training set. To compare the effectiveness of the two models, SOC estimation based on electrical characteristics (voltage and current) using DNN was also carried out. Compared to the electrical model, the nonelectrical model has a lower SOC estimation accuracy of roughly 1.2%.

The estimation of SOC and SOH and capacity prediction have been achieved based on the temperature and strain obtained by FBG sensors. The application of FBG sensors in monitoring lithium-ion battery safety is listed in Table 2.

In addition, the electrochemical stability of the liquid electrolyte state is important for the SOH of lithium-ion batteries. The monitoring of electrolyte RI is important to monitor electrolyte decomposition and turbidity. The relationship between temperature, electrolyte RI, and battery capacity loss has been proven [5]. Along with the artificial intelligence algorithm, these two observables can be used to predict battery aging and serve as useful inputs for the BMS.

### 5.2. Safety Warning for Lithium-Ion Batteries

The monitored data of temperature, strain, and pressure can be used for safety warnings to prevent accidents such as overheating explosions, electrode cracking and battery bulges, and gas-release events.

To avoid a battery pack overheating, lithium-ion battery temperatures must be closely monitored [55,94]. The construction of sophisticated heating/cooling systems with the necessary temperature-triggered thresholds to prevent disastrous thermal events may be aided by temperature monitoring in combination with the heat that the battery has stored. To avoid electrode cracking and battery bulges, it is crucial to detect strain on the electrode and the external surface. It has been proven that, on average, the maximum gas pressure arrives 290 s before the peak temperature [26]. Lithium-ion battery gas pressure can therefore act as an early warning sign of battery overheating. An efficient pre-warning before gas release occurrences is provided by the real-time monitoring of the internal pressure [61].

It should be emphasized that early warning signals with a certain safety margin should be put forward for smart battery safety, i.e., the safety alarm should be given before the batteries completely enter the thermal runaway state and the batteries are still in the critical stable state, rather than the immediate feedback of thermal runaway. Otherwise, it will significantly increase the possible safety risks and diminish the significance of the warning. The mechanical mode of smart batteries, which is based on a multi-parameter FBG sensor, can deliver a warning several to tens of seconds earlier than traditional approaches to tracking battery temperature [55].

## 6. Conclusions and Future Perspectives

In the last decade, FBG sensors for lithium-ion battery safety monitoring have experienced rapid development. They have been successfully applied in the estimation of SOC and SOH, which can assist the BMS in accurately controlling the operating status of each cell. Compared with traditional electronic sensors, FBG sensors have excellent characteristics, such as microstructure, anti-electromagnetic interference, and distributed measurement capabilities. They can monitor the internal parameters of battery cells, which cannot be realized by the current BMS. Many parameters of lithium-ion batteries have been successfully monitored by FBG sensors, including temperature, strain, pressure, and electrolyte refractive index (RI). The observed data can be utilized to estimate SOC and SOH and provide lithium-ion battery safety warnings, which can provide enough information in real-time to avoid battery accidents such as thermal runways. However, most test results are still in the laboratory research stage, as reported in the literature. Before FBG sensors can be employed in lithium-ion batteries for practical applications, there are still a few obstacles to overcome.

The first is the efficient application of FBG sensors. It is challenging to implant FBG sensors inside lithium-ion batteries due to their narrow internal area. The usual service of batteries could be significantly impacted by the implanted FBG sensors. The protection structure is necessary because the bare optical fiber is vulnerable. Lithium-ion batteries electrolyte exhibits strong chemical corrosion, which could shorten the lifespan of FBG sensors. It is essential to guarantee the lithium-ion batteries proper operation and the long-term stability of the FBG sensors. The corrosion-resistant protective film is useful for bare FBG. We can plate the grating area with gold to protect the bare FBG from corrosion.

The second is the calibration of FBGs implanted in lithium-ion batteries. FBG sensors can be calibrated before they implanted into lithium-ion batteries. However, the extremely complex internal environment of lithium-ion batteries may affect the accuracy of the calibration results. Additionally, there are significant differences in the internal makeup of various lithium-ion batteries. As a result, it is challenging to calibrate FBG sensors accurately with various lithium-ion battery types. To solve this problem, we can insert a sealed standard platinum resistor inside the different lithium-ion batteries (only used for calibration) and calibrate the FBG regularly.

The third is the effective application of monitored data. Multiple parameters of lithium-ion batteries have been obtained by FBG sensors. However, the monitored data have not been sufficiently used for the safety monitoring of lithium-ion batteries. The estimation algorithms of SOC and SOH based on the monitored data need to be improved. The safety warning system is still not perfect, and a fast and accurate response safety warning system needs to be built based on the monitored data. To achieve the goal, we can establish the mathematical models of SOC and SOH estimation and safety early warning systems based on multiple parameters, then optimize the models according to the experimental test results.

## Figures and Tables

**Figure 1 sensors-23-05609-f001:**
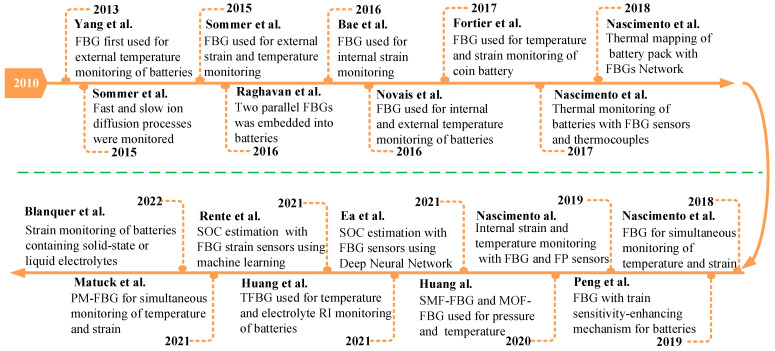
Timeline of the development of the FBG sensor in lithium-ion batteries [30,49,50,51,52,53,54,55,56,57,58,59,60,61,62,63,64,65].

**Figure 2 sensors-23-05609-f002:**
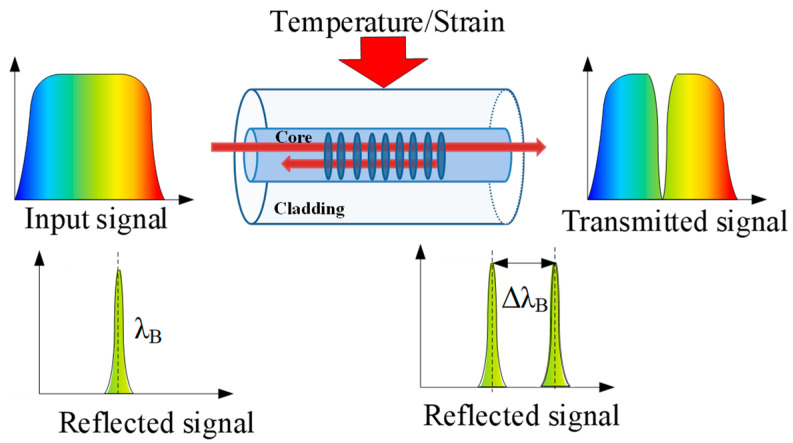
Schematic illustration of the FBG structure and sensing principle [68].

**Figure 3 sensors-23-05609-f003:**
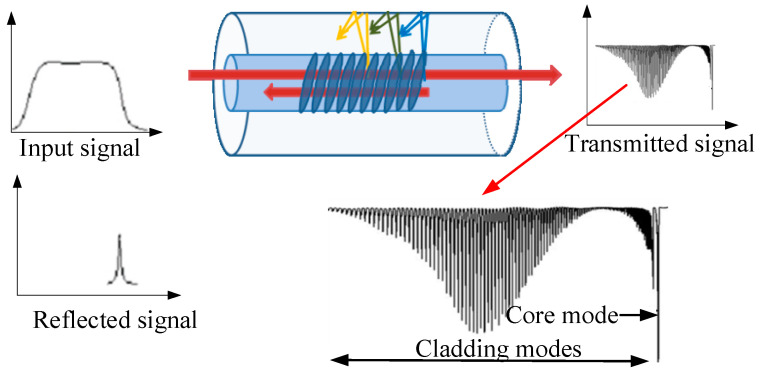
Schematic illustration of the TFBG [73].

**Figure 4 sensors-23-05609-f004:**
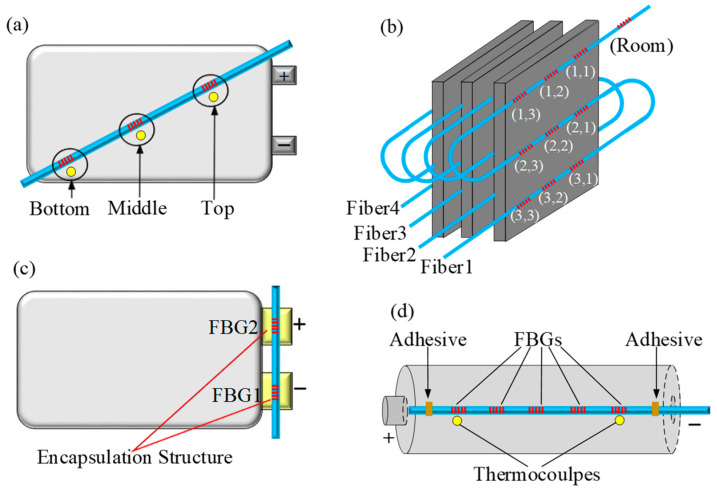
External temperature monitoring of lithium-ion batteries (**a**) with three FBGs [56]; (**b**) with 37 FBGs [57]; (**c**) with an encapsulation structure [58]; (**d**) with five FBGs [59].

**Figure 5 sensors-23-05609-f005:**
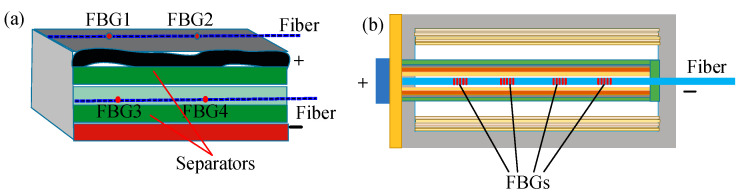
Internal temperature monitoring of lithium-ion batteries. (**a**) pouch lithium-ion batteries [54]; (**b**) cylindrical lithium-ion batteries [80].

**Figure 6 sensors-23-05609-f006:**
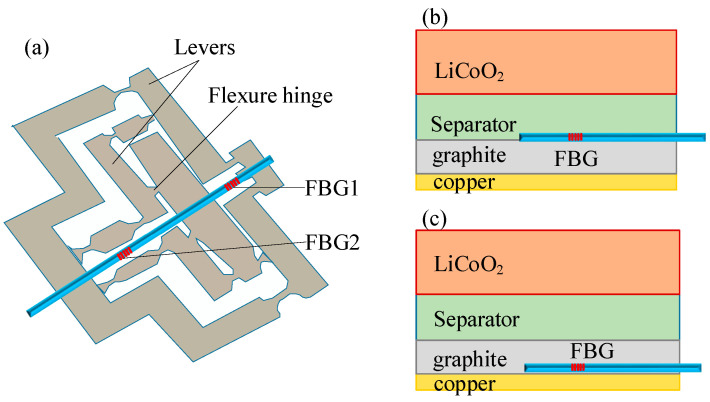
External and internal strain monitoring of lithium-ion batteries. (**a**) Sensitivity-enhanced structure for external strain monitoring [59]; (**b**,**c**) two configurations of FBG for internal strain monitoring [53].

**Figure 7 sensors-23-05609-f007:**
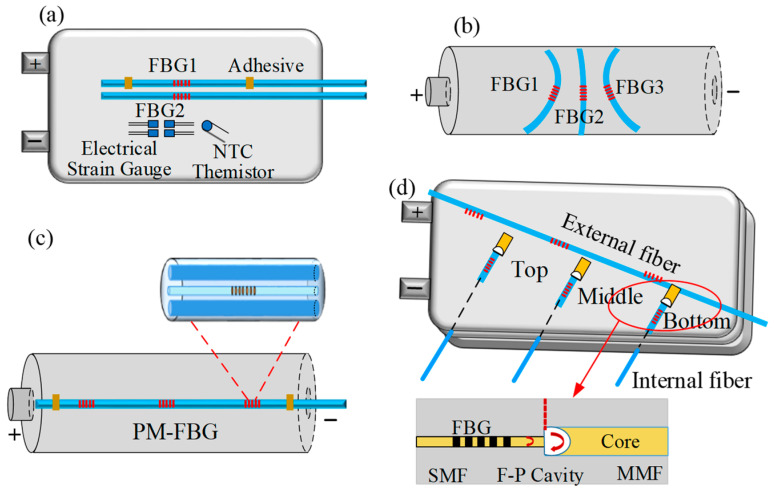
Simultaneous monitoring of temperature and strain scheme. (**a**) Parallel reference FBG [50]; (**b**) Tilted fixed three FBGs [63]; (**c**) Polarization-Maintaining FBG [64]; (**d**) FBG combined with F-P Interferometer [60].

**Figure 8 sensors-23-05609-f008:**
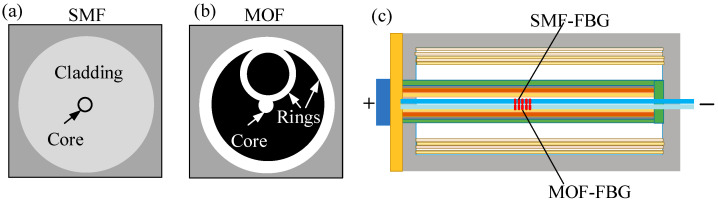
Simultaneous monitoring of temperature and pressure [61]. (**a**,**b**) cross-section of SMF and MOF; (**c**) SMF-FBG and MOF-FBG implanted into lithium-ion batteries.

**Table 1 sensors-23-05609-t001:** Summary of some FBG sensors developed to monitor different parameters of lithium-ion batteries: temperature, strain, pressure, and electrolyte RI.

Sensor Type	Measured Parameter	Sensitivity	Resolution/Accuracy	Battery Type	Year	Ref.
FBG	external temperature	9.97 pm/°C	2.06 °C	cylindrical	2021	[79]
	external temperature	10.3 pm/°C	0.1 °C	cylindrical	2018	[57]
	external temperature	9.24 pm/°C	0.11 °C	pouch	2017	[56]
	external temperature	12 pm/°C	0.08 °C	pouch	2021	[78]
	external temperature	10 pm/°C	0.1 °C	coin	2013	[49]
	internal Temperature	11 pm/°C	-	cylindrical	2018	[80]
	external and internal temperature	10.27 pm/°C	0.1 °C	pouch	2016	[54]
	external strain	11.55 pm/uε	0.09 uε	pouch	2019	[59]
	internal strain	-	-	Swagelok	2021	[81]
Parallel reference FBG	external temperature and strain	8.04 pm/°C, 1.2 pm/uε	0.12 °C, 0.83 uε	pouch	2018	[58]
Titled Fixed three FBGs	external temperature and strain	21 pm/°C, 1.0 pm/uε	-	cylindrical	2021	[63]
PM-FBG	external temperature and strain	23.7 pm/°C, 1.2 pm/uε	0.04°C, 0.83 uε	cylindrical	2022	[64]
FBG+FPI	internal temperature and strain	40 pm/°C, 2.2 pm/uε	0.1 °C, 0.1 uε	pouch	2019	[60]
SMF-FBG+MOF-FBG	Internal temperature and pressure	10 pm/°C, −7.2 pm/bar	0.1 °C, 0.14 bar	cylindrical	2021	[61]
TFBG	Internal temperature and electrolyte RI	−18 nm/RIU, 10.1 pm/°C	6 × 10^−5^ RIU	cylindrical	2021	[30]

**Table 2 sensors-23-05609-t002:** Application of FBG sensors in monitoring lithium-ion battery safety.

Application	Algorithm	Accuracy	Battery Type	Year	Ref.
SOC	DWT	2.5%	Pouch	2017	[92]
	DWT	2%	Cylindrical	2021	[63]
	DNN	1.2%	Pouch	2021	[62]
SOH	DWT	1.1%	Pouch	2017	[92]
Capacity Prediction	Kalman filtering	2%	Pouch	2017	[92]

## Data Availability

Not applicable.

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
