# Peer review of "Recent Progress in Lithium-Ion Battery Safety Monitoring Based on Fiber Bragg Grating Sensors"

_sensors, 2023, doi:10.3390/s23125609_

Round 1
Reviewer 1 Report
In this paper, the authors reviewed lithium-ion battery safety monitoring based on FBG sensors. The principles and sensing performance of FBG sensors are described. The work is useful for this area and can be published after following revisions:
1) It is better to give a timeline of the developing of FBG in LIB.
2) The limitation of FBG could be summarized and discussed.
3) The format and language could be improved.
4) Related references may be cited, such as DOI:10.1021/acsami.1c12589.
Acceptable.
Reviewer 2 Report
Please find enclosed a revision of the manuscript “Recent Progress in Lithium-ion Battery Safety Monitoring Based on Fiber Bragg Grating Sensors”. Authors: Dongying Chen, Qiang Zhao, Yi Zheng, Yuzhe Xu, Yonghua Chen, Jiasheng Ni and Yong ZhaoWe. This manuscript “Recent Progress in Lithium-ion Battery Safety Monitoring Based on Fiber Bragg Grating Sensors” provides a concise overview of the importance of real-time safety monitoring for lithium-ion batteries, given their frequent safety incidents despite their widespread use in various fields. The document specifically focuses on the application of fiber Bragg grating (FBG) sensors for lithium-ion battery safety monitoring. The main objectives are clearly stated: to describe the principles and sensing performance of FBG sensors, review the monitoring techniques using single-parameter and dual-parameter approaches, summarize the current application state of monitored data in lithium-ion batteries, provide an overview of recent developments in FBG sensors for lithium-ion batteries, and discuss future trends in this field. The manuscript offers readers relevant information, and I consider it eligible for publication after the following changes:
1.The introduction provides a general overview of the importance of lithium-ion batteries in energy storage devices and the need for effective battery management systems (BMS). However, there are several areas where the introduction can be improved to enhance clarity and provide a stronger foundation for the subsequent sections. Here are some recommendations:
a.-Begin the introduction by clearly stating the purpose of the review and the specific objectives it aims to address. This will provide a concise and focused introduction that guides the reader. For example:
"In this review, we aim to provide a comprehensive analysis of the safety monitoring of lithium-ion batteries based on fiber Bragg grating (FBG) sensors. Our objectives are to explore the potential of FBG sensors in monitoring various parameters, such as temperature, strain, and gas pressure, to enhance the safety, state of charge (SOC), and state of health (SOH) estimation of lithium-ion batteries."
b.-The first paragraph can be restructured to provide a more logical flow of information. Begin by introducing the importance of lithium-ion batteries and their advantages, followed by the limitations of current battery management systems. Then, introduce the use of FBG sensors as a promising solution.
c.- The transition between ideas should be improved to ensure a smooth flow of information. Instead of simply stating the sections, provide a brief summary of what each section will cover. For example:
"To provide a comprehensive understanding of FBG-based safety monitoring in lithium-ion batteries, we have organized this review as follows: Section 2 will provide an overview of the working principles of fiber Bragg grating. Sections 3 and 4 will discuss the single-parameter and dual-parameter monitoring techniques employed in lithium-ion batteries. Furthermore, in Section 5, we will explore the utilization of monitored data. Finally, Section 6”
2.Provide a more comprehensive explanation of the sensing principle of TFBG sensors, focusing on their sensitivity to the surrounding refractive index (RI) and their ability to decouple multiple parameters. Lines 142-151.
3.- A brief introduction that summarizes the focus on each section is welcome. This will provide context and enhance the understanding of the subsequent descriptions. Specially section 3 and 4.
4.- The Conclusions section must provide a comprehensive overview of the achievements, challenges, and potential future directions in the field of FBG sensors for lithium-ion battery safety monitoring, promoting a deeper understanding of the topic for readers. To improve the Conclusions section, I recommend the following revisions:
a.-Summarize the significant advancements and achievements in the field of FBG sensors for lithium-ion battery safety monitoring over the past decade. Emphasize the benefits of FBG sensors, such as their microstructure, anti-electromagnetic interference capabilities, and distributed measurement capabilities.
b.-Clearly state the parameters that have been successfully monitored using FBG sensors, including temperature, strain, pressure, and electrolyte refractive index (RI). This will help readers understand the breadth of information that can be obtained through FBG sensors.
c.-Highlight the potential applications of the monitored data, specifically in estimating State of Charge (SOC) and State of Health (SOH) of lithium-ion batteries and providing safety warnings. Emphasize the significance of these applications in improving battery performance and preventing safety incidents.
No comments
Reviewer 3 Report
In this manuscript, the authors have conducted a review on the safety monitoring of lithium-ion batteries based on Fiber Bragg Grating (FBG) technology. The review covers various aspects, including the working principles of FBG, multi-parameter monitoring (such as temperature, strain, pressure, and electrolyte RI), utilization of monitored data, and the potential perspectives of FBG sensors in battery management systems (BMS). The findings of this review demonstrate the promising application potential of FBG in enhancing the safety of BMS. Therefore, I recommend its publication with minor revisions.
1. The authors should provide evidence of the service value of FBG sensors by considering their integration into the overall battery working process.
2. The authors should provide an introduction to the materials and assembly units used in advanced FBG sensors, and address manufacturing issues and cost-related concerns.
3. I suggest that the authors propose feasible solutions for each of the obstacles mentioned in the perspectives section.
4. I suggest that the authors discuss and compare the service value of FBG sensors in different battery forms, such as coin cells, pouch cells, solid-state batteries, wearable batteries, and others.
5. I suggest that the authors cite recently published research articles related to this topic and establish relevant evaluation criteria for advanced FBG sensors.
Reviewer 4 Report
Accept
Minor editing of English language required
Author Response
Dear reviewer:
Thank you for your comments concerning our manuscript entitled “Recent Progress in Lithium-ion Battery Safety Monitoring Based on Fiber Bragg Grating Sensors” (ID: sensors-2436607). We gratefully appreciate for your comment. We have carefully checked and improved English language in the revised manuscript. Thanks again for your valuable comment.
Yours sincerely,
Dongying Chen, Qiang Zhao, Yi Zheng, Yuzhe Xu, Yonghua Che, Jiasheng Ni, Yong Zhao
Corresponding author:
Name: Qiang Zhao
Email: zhaoqiang@qlu.edu.cn
Reviewer 5 Report
This review presents a interesting topic which can attract great attention of researchers and comunity. However, its construction and design are not good. Thus it should be significantly revised according to following major comments.
1. This is a review. All figures are not original, they are coppied or modified from specific references. Please add the appropriate reference in the figure caption.
2. This review only concentrates on discussing the sensing principle and construction of FBG sensors. I think that novel materials used for fabrication of FBG sensors are also an important topic. I recommend that the authors should insert this content.
3. The state-of-the-art application of FBG sensors in monitoring lithium-ion battery safety is neccessary for providing recent trends. It should be summaried in a table.
4. The introduction is long, but it is not linked and compact. Please carefully revise it in term of emphasizing the importance of FBG sensors in monitoring lithium-ion battery safety and the problems as well as drawbacks of the current technologies.
5. In the sensing principle of FBG sensors, please indicate and emphasize its important parameters
N/A
Round 2
Reviewer 5 Report
The manuscript can be accepted for publication in the current form
N/A